# Is Cognition consistent with Perception?
# ASSESSING AND MITIGATING MULTIMODAL KNOWLEDGE CONFLICTS IN DOCUMENT UNDERSTANDING

## ABSTRACT

Multimodal large language models (MLLMs) have shown impressive capabilities in document understanding, a rapidly growing research area with significant industrial demand in recent years. As a multimodal task, document understanding requires models to possess both perceptual and cognitive abilities. However, current MLLMs often face conflicts between perception and cognition. Taking a document VQA task (cognition) as an example, an MLLM might generate answers that do not match the corresponding visual content identified by its OCR (perception). This conflict suggests that the MLLM might struggle to establish an intrinsic connection between the information it "sees" and what it "understands." Such conflicts challenge the intuitive notion that cognition is consistent with perception, hindering the performance and explainability of MLLMs. In this paper, we define the conflicts between cognition and perception as *Cognition and Perception (C&P) knowledge conflicts*, a form of multimodal knowledge conflicts, and systematically assess them with a focus on document understanding. Our analysis reveals that even GPT-4o, a leading MLLM, achieves only 68.6% C&P consistency. To mitigate the C&P knowledge conflicts, we propose a novel method called *Multimodal Knowledge Consistency Fine-tuning*. This method first ensures task-specific consistency and then connects the cognitive and perceptual knowledge. Our method significantly reduces C&P knowledge conflicts across all tested MLLMs and enhances their performance in both cognitive and perceptual tasks in most scenarios.

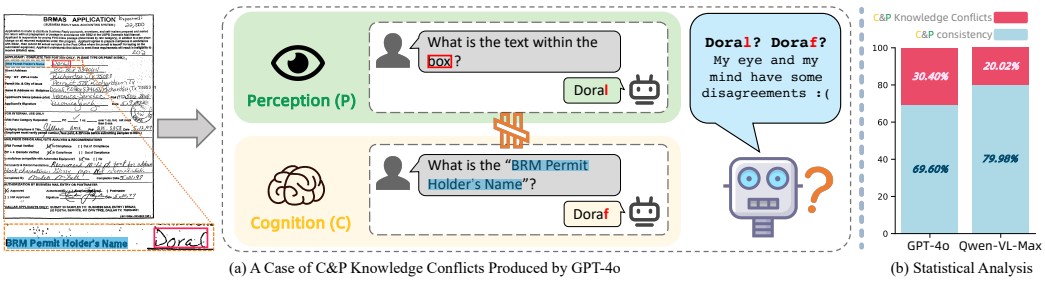

(a) A Case of C&P Knowledge Conflicts Produced by GPT-4o      (b) Statistical Analysis

Figure 1: *a*: GPT-4o generates a VQA (cognition) answer that conflicts with the corresponding visual content identified by its OCR (perception). We refer to these multimodal knowledge conflicts in MLLMs as *Cognition and Perception (C&P) knowledge conflicts*. *b*: Statistical analysis of C&P knowledge conflicts in leading MLLMs (Section 3).

## 1 INTRODUCTION

In recent years, multimodal large language models (MLLMs) (gpt, 2023; Team et al., 2023; gpt, 2024; Chen et al., 2024; Bai et al., 2023; Ye et al., 2024; Li et al., 2024a) have witnessed rapid development and have demonstrated remarkable capabilities across a wide range of multimodal tasks (Antol et al., 2015; Mathew et al., 2021; Hossain et al., 2019). Particularly in the field of

document understanding (Cui et al., 2021; Xu et al., 2020; 2021; Huang et al., 2022; Gu et al., 2022; Luo et al., 2023), which has high academic and industrial value, significant research efforts with MLLMs have been made (Zhang et al., 2023a; Ye et al., 2023a;b; Luo et al., 2024; Wang et al., 2023; Hu et al., 2024), yielding promising results.

As a multimodal task, document understanding requires models to accurately perceive visual content (perception) and then generate coherent responses (cognition) based on that perception. However, current MLLMs often face conflicts between perception and cognition. For example in Figure 1 (a), GPT-4o (gpt, 2024) recognizes the text in a certain region of an image as "Doral" through its OCR capability (perception) but responds to a related information extraction question with the text "Doraf" (cognition). This conflict suggests that the GPT-4o might struggle to establish an intrinsic connection between what it "sees" and what it "understands." Statistical analysis further underscores this issue, as Figure 1 (b) shows, with leading MLLMs like GPT-4o and Qwen-VL-Max (Bai et al., 2023) achieving 69.60% and 79.98% consistency between perception and cognition (Section 3).

In this paper, we define intrinsic conflicts between cognitive knowledge and perceptual knowledge within MLLMs, which result in inconsistencies in responses related to cognition and perception, as *Cognition and Perception (C&P) knowledge conflicts* (Section 2.1). C&P knowledge conflicts serve as a critical factor undermining the explainability of MLLM responses, as these conflicts challenge the intuitive notion that cognition is consistent with perception. Unlike previous research on multimodal knowledge conflicts (e.g., hallucination) (Zhai et al., 2024; Li et al., 2023; Guan et al., 2024; Liu et al., 2023a), which focuses solely on conflicts within either cognition or perception, we highlight, for the first time, the conflicts that arise between the two.

We systematically assess C&P knowledge conflicts in the current five MLLMs (Section 3), focusing on document understanding. Here, the cognitive task is document-related VQA, while the perceptual task is OCR. The experimental results show significant C&P knowledge conflicts in current MLLMs, underscoring the need to mitigate these conflicts. To address this, a novel method called *Multimodal Knowledge Consistency Fine-tuning* is introduced, which includes three fine-tuning tasks (Section 4). Specifically, motivated by the Generator-Validator (GV) framework (Li et al., 2024b), we conduct two task-specific fine-tuning tasks: the *Cognition Consistency* task and the *Perception Consistency* task. The purpose of these two tasks is based on our belief that ensuring C&P consistency starts with maintaining task-specific consistency. Furthermore, to establish an inner connection between cognitive and perceptual knowledge, the third fine-tuning task is designed: the *C&P Connector* task.

Comprehensive experiments are conducted on three open-source MLLMs across two series and two parameter sizes. The results indicate that multimodal knowledge consistency fine-tuning significantly improves C&P consistency, with all three MLLMs achieving at least a 34% improvement (Section 5.2). Moreover, in most scenarios, our method also enhances MLLM performance in both cognitive and perceptual tasks (Section 5.4).

Our main contributions are as follows:

- To the best of our knowledge, we are the first to identify and introduce the concept of Cognition and Perception knowledge conflicts, a form of multimodal knowledge conflicts, in MLLMs.

- A systematic evaluation is conducted on current MLLMs to assess the Cognition and Perception knowledge conflicts in document understanding, showing that such conflicts are commonly present in current MLLMs.

- A novel method called Multimodal Knowledge Consistency Fine-tuning is introduced to mitigate the C&P knowledge conflicts in current MLLMs. Extensive experiments on six public document understanding benchmarks in three MLLMs demonstrate the effectiveness of the proposed method.

## 2 PROBLEM STATEMENT

### 2.1 THE DEFINITION OF COGNITION AND PERCEPTION KNOWLEDGE CONFLICTS

For a given MLLM $f(\cdot)$, an image $x_I$, and a pair of queries consisting of a cognitive task query $x_C$ and a perceptual query $x_P$, we denote the ground truth for this pair as $y$. The MLLM's responses for

Table 1: Data statistics for C&P knowledge conflicts evaluation. The number of evaluation samples, i.e., cognitive (VQA) query and perceptual (OCR) query $(x_C, x_P)$ pairs, along with the corresponding images for each dataset.

|  | DocVQA | DeepForm | KLC | FUNSD | ChartQA | WTQ |
|---|---|---|---|---|---|---|
| # Evaluation Samples | 4440 | 687 | 1212 | 422 | 1532 | 2391 |
| # Images | 1244 | 266 | 563 | 46 | 1198 | 379 |

cognitive and perceptual tasks are represented as $y_C = f(x_C, x_I)$ and $y_P = f(x_P, x_I)$, respectively. Let $\mathcal{K}$ represent the complete knowledge embedded in the MLLM $f(\cdot)$. The subset of $\mathcal{K}$ used by $f(\cdot)$ to generate the cognitive response $y_C$ is referred to as *cognitive knowledge* and is denoted by $\mathcal{K}_C$, while the subset used for the perceptual response is termed *perceptual knowledge* and is denoted by $\mathcal{K}_P$.

Conflicts arise between $\mathcal{K}_C$ and $\mathcal{K}_P$, referred to as *Cognition and Perception (C&P) knowledge conflicts*, resulting in $y_C$ and $y_P$ being inconsistent (i.e., $\delta(y_C, y_P) = 0$). It is important to note that C&P knowledge conflicts do not consider whether $y_C = y$ or $y_P = y$. To quantify the severity of these conflicts, we introduce C&P consistency. Let $N$ denote the number of $(y_C, y_P)$ pairs, with the C&P consistency calculated as follows:

$$\text{C\&P Consistency} = \frac{\sum_{i=1}^{N} \delta(y_{C_i}, y_{P_i})}{N}. \tag{1}$$

In this paper, we focus on document understanding, where given a text $GT$ within $x_I$ bounded by $Box$, $x_C$ is a VQA query using $GT$ as the answer, and $x_P$ is an OCR query operating solely within $Box$. In practice, $Box$ may contain additional text besides $GT$. Consequently, C&P knowledge conflicts occur when $y_P$ does not fully contain $y_C$. The $\delta(y_C, y_P)$ can be specifically defined as follows:

$$\delta(y_C, y_P) = \begin{cases} 1, & \text{if } y_C \subseteq y_P \\ 0, & \text{if } y_C \nsubseteq y_P \end{cases} \tag{2}$$

## 2.2 TASKS

As shown in Table 1, we consider six document understanding datasets to assess C&P knowledge conflicts, categorized into the following four tasks:

**Document QA.** DocVQA (Mathew et al., 2021) contains 50k question-answer pairs based on 12k document images from the UCSF Industry Documents Library.

**Document IE.** DeepForm (Svetlichnaya, 2020), Kleister Charity (KLC) (Stanisław et al., 2021), and FUNSD (Jaume et al., 2019) are three Information Extraction datasets. DeepForm consists of 1.1k documents related to election spending, while KLC includes 2.7k documents from published charity organization reports. FUNSD contains 0.2k document images from the RVL-CDIP dataset (Harley et al., 2015). The annotations for DeepForm, KLC, and FUNSD are transformed into a question-answer format, with DeepForm and KLC following Hu et al. (2024), and FUNSD following Luo et al. (2024).

**Chart QA.** ChartQA (Masry et al., 2022) compiles a diverse range of topics and chart types from four primary sources: Statista (statista.com), The Pew Research Center (pewresearch.org), Our World in Data (ourworldindata.org), and the OECD (oecd.org). In total, the dataset includes 21k chart images and 32k question-answer pairs.

**Table QA.** WikiTableQuestions (WTQ) (Pasupat & Liang, 2015) dataset consists of 2.1k table images from Wikipedia, annotated with 23k question-answer pairs.

Notably, OCR annotations are required in Section 2.3. For the DocVQA dataset, the official OCR annotations are used, while the other datasets use OCR annotations produced by Duguang OCR[1].

---

[1] https://duguang.aliyun.com/

## 2.3 THE CONSTRUCTION OF EVALUATION SAMPLES

To calculate C&P consistency, we construct several pairs of cognitive (VQA) query and perceptual (OCR) query, i.e., $(x_C, x_P)$, with each pair using the same ground truth $GT$ from the image $x_I$. The process, as shown in Figure 2, is as follows:

Since each image is accompanied by original question-answering annotations (Section 2.2), given an image $x_I$ with its QA annotation $(Q, A)$, we assign $GT = A$ and $x_C = Q$. $x_P$ is constructed in QA format with the template $Temp_P =$ "What is the text within {Box}?", where $Box$ is the bounding box containing $GT$ in $x_I$, i.e., $x_P = Temp_P(Box)$. Since the $Box$ annotations are not provided, the $Box$ is identified by searching the OCR annotations of $x_I$ for $A$.

However, not all $(Q, A)$ pairs can be used to construct $(x_C, x_P)$ pairs due to some $A$ not appearing in the OCR annotations, which can be categorized into two scenarios: (1) According to the definition in Section 2.1, the questions must pertain to the text in the image. However, certain questions, such as those related to comparisons or yes/no answers, do not directly reference the text. To address this, we apply keyword-based filtering to exclude such QA pairs. (2) Since the OCR annotations are generated by third-party OCR engines, some answers may not be present in the OCR annotations due to issues like OCR errors. These QA pairs are also filtered out.

Figure 2: A specific example illustrates the process of evaluation sample construction. All mathematical symbols in the figure are consistent with those in Section 2.3. Corresponding relationships are represented using the same colors for clarity.

The evaluation samples are constructed on the test sets of all datasets in Section 2.2, as shown in Table 1, which lists the number of $(x_C, x_P)$ pairs along with their corresponding images. Additionally, there are minor differences in $x_P$ between closed-source and open-source MLLMs. Since detailed information about the bounding box input format for closed-source models is not publicly available, we draw a prominent red bounding box in $x_I$ at the location of $Box$, inspired by Set-of-Mark prompting (Yang et al., 2023). For open-source models, we follow the bounding box input format outlined in their papers (Bai et al., 2023; Chen et al., 2024) to construct $x_P$.

# 3 THE COGNITION AND PERCEPTION KNOWLEDGE CONFLICTS IN CURRENT MLLMS

Two closed-source and three open-source MLLMs are evaluated. The closed-source models, GPT-4o[2] (gpt, 2024) and Qwen-VL-Max[3] (Bai et al., 2023), are both well-regarded in the community. These models are evaluated using their publicly available APIs, with all tests conducted in September 2024. The open-source models include Qwen-VL-Chat-7b[4] (Bai et al., 2023), InternVL2-2b[5] (Chen et al., 2024), and InternVL2-8b[6] (Chen et al., 2024), which differ in size and architecture. We use weights available on Huggingface (Wolf et al., 2020), and the evaluation is performed on an Nvidia A100 GPU.

Table 2 shows the evaluation results. Overall, closed-source models have higher C&P consistency compared to open-source models. Qwen-VL-Max achieves the highest C&P consistency at 79.98%, followed by GPT-4o at 68.60%. Among the open-source models, Qwen-VL-Chat demonstrates the

---

[2] https://platform.openai.com
[3] https://www.alibabacloud.com
[4] https://huggingface.co/Qwen/Qwen-VL-Chat
[5] https://huggingface.co/OpenGVLab/InternVL2-2B
[6] https://huggingface.co/OpenGVLab/InternVL2-8B

Table 2: C&P Knowledge Conflicts in Current MLLMs. All values represent C&P consistency as a percentage (%). Bold indicates the best results among closed-source MLLMs, while underlined indicates the best results among open-source MLLMs. The average results are the micro-averages of all datasets.

|  | DocVQA | DeepForm | KLC | FUNSD | ChartQA | WTQ | Average |
|---|---|---|---|---|---|---|---|
| GPT-4o | 77.91 | 23.07 | 81.68 | 77.73 | 68.47 | 57.07 | 68.60 |
| Qwen-VL-Max | **87.20** | **43.39** | **88.06** | **81.91** | **82.69** | **70.54** | **79.98** |
| Qwen-VL-Chat | 20.82 | 5.240 | 37.87 | 7.264 | 21.64 | 8.672 | 19.41 |
| InternVL2-2b | 13.92 | 1.456 | 18.48 | 7.506 | 9.107 | 10.30 | 12.09 |
| InternVL2-8b | 20.47 | 3.202 | 30.53 | 11.14 | 9.558 | 9.214 | 16.87 |

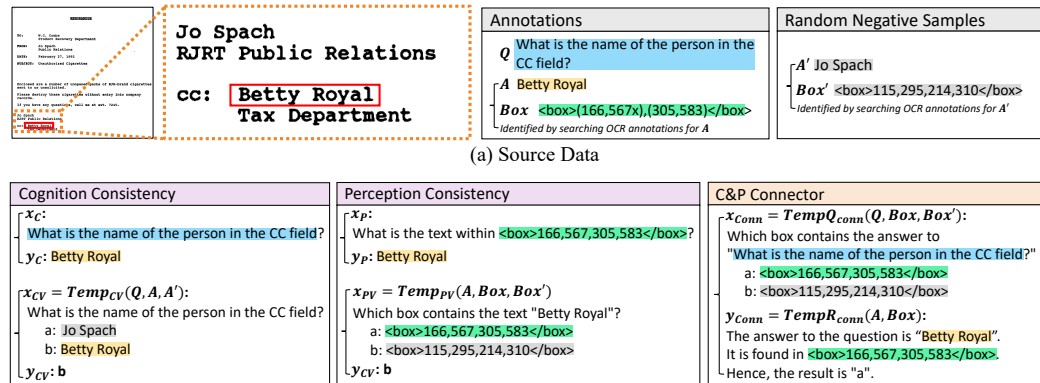

(a) Source Data

(b) Multimodal Knowledge Consistency Fine-tuning Tasks

Figure 3: An example illustrates the source data *(a)* and its corresponding *Multimodal Knowledge Consistency Fine-tuning* sample *(b)*. Multimodal knowledge consistency fine-tuning consists of *Cognition Consistency* task and *Perception Consistency* task for task-specific consistency, while the *C&P connector* task connects cognitive and perceptual knowledge. All mathematical symbols in the figure are consistent with those in Section 4. Corresponding relationships are represented using the same colors for clarity.

best C&P consistency, though it remains below 20%. Additionally, we observe that the size of MLLM parameters affects C&P consistency, as InternVL2-8b performs better than InternVL2-2b. Furthermore, C&P consistency varies across datasets. For instance, all MLLMs perform best on DocVQA but perform worst on DeepForm. This may be related to the layout of document images in DeepForm, which typically contain a large amount of small text.

## 4 MULTIMODAL KNOWLEDGE CONSISTENCY FINE-TUNING

Table 2 demonstrates that even leading MLLMs face C&P knowledge conflicts, which negatively affect explainability. To resolve these conflicts, we introduce a novel method called *Multimodal Knowledge Consistency Fine-tuning*, as shown in Figure 3.

We suggest that ensuring C&P consistency starts with maintaining task-specific consistency, meaning consistency within cognitive and perceptual tasks. Li et al. (2024b) introduces the Generator-Validator (GV) fine-tuning framework, which ensures task consistency by generating mutually validating queries. Building on this framework, we construct validation queries $x_{CV}$ and $x_{PV}$ for the cognitive query $x_C$ and the perceptual query $x_P$, respectively, referring to $((x_C, y_C), (x_{CV}, y_{CV}))$ as the *Cognition Consistency* task and $((x_P, y_P), (x_{PV}, y_{PV}))$ as the *Perception Consistency* task.

The $x_C$ and $x_P$ are constructed following Section 2.3. Let an image $x_I$, its QA annotation $(Q, A)$, and the bounding box $Box$ of $A$ on $x_I$ be given. The validation queries $x_{CV}$ and $x_{PV}$ are framed as two-option questions, using templates $Temp_{CV}$ and $Temp_{PV}$, respectively. Specifically, $x_{CV} =$

$Temp_{CV}(Q, A, A')$, where $A'$ is a negative sample, randomly selected from the text in $x_I$ excluding $A$. Similarly, $x_{PV} = Temp_{PV}(A, Box, Box')$, where $Box'$ is a bounding box randomly sampled from $x_I$, excluding $Box$. It is important to note that the construction of $(x_P, x_{PV})$ is independent of $A$, which means they can be generated using all the text and their bounding boxes across the entire image. Additionally, the order of the options is randomly shuffled to ensure balanced data.

Additionally, to establish a connection between cognitive and perceptual tasks, we designed the *C&P Connector*. Let $x_{Conn}$ and $y_{Conn}$ represent the query and response of the C&P Connector, respectively. Formally, $x_{Conn} = TempQ_{Conn}(Q, Box, Box')$ and $y_{Conn} = TempR_{Conn}(A, Box)$, where $TempQ_{Conn}$ is the query template (a two-option question), and $TempR_{Conn}$ is the response template. The goal is to utilize the query and response from the C&P Connector to link $Q$, $A$, and $Box$, thus creating a bridge between the cognitive and perceptual tasks, and reducing knowledge conflicts.

For the specific training strategy, we implement a three-stage approach. Given $N$ pairs of $(Q, A)$, the details are as follows:

- **Stage 1**: Perception Consistency, denoted as $\mathcal{X}_{s1} = \{((x_{P_i}, y_{P_i}), (x_{PV_i}, y_{PV_i})) \mid i = 0, 1, \ldots, M\}$. To enhance data efficiency, we use all text and their corresponding bounding boxes from the entire image, resulting in $M \gg N$.

- **Stage 2**: Cognition Consistency, denoted as $\mathcal{X}_{s2} = \{((x_{C_i}, y_{C_i}), (x_{CV_i}, y_{CV_i})) \mid i = 0, 1, \ldots, N\}$.

- **Stage 3**: Establishing Connections, denoted as $\mathcal{X}_{s3} = \{(x_{Conn_i}, y_{Conn_i}) \mid i = 0, 1, \ldots, W\} \cup \mathcal{X}_{s1}^{\text{sub}} \cup \mathcal{X}_{s2}^{\text{sub}}$. As explained in Section 2.3, some $(Q, A)$ pairs cannot be used to construct the C&P Connector, resulting in $W < N$. Additionally, we incorporate a small amount of data from previous stages to maintain model performance.

# 5 EXPERIMENT

## 5.1 IMPLEMENTATION

We construct the training data using the training sets from the six datasets mentioned in Section 2.2. To simplify DeepForm and KLC, their Cognition Consistency training data are constructed solely from the QA pairs that pass the filtering process in Section 2.3. Following Section 4, the training data for Stage 1, Stage 2, and Stage 3 contain 2189k, 176k, and 146k training samples, respectively.

For the multimodal knowledge consistency fine-tuning experiment, we focus on three open-source MLLMs (Section 3): Qwen-VL-Chat-7b, InternVL2-2b, and InternVL2-8b. All models are trained with a learning rate of 1e-5 and a batch size of 128, while other hyperparameters remain at their default settings. We freeze the visual encoder and optimize only the language model. Each model trains for 1 epoch using 8 Nvidia A100 GPUs.

## 5.2 MAIN RESULTS

The evaluation is performed on the dataset constructed in Section 2.3. In addition to C&P Consistency, we also report *Cognitive Task Consistency* and *Perceptual Task Consistency*. Following Li et al. (2024b), cognitive task consistency quantifies the percentage of cases where $y_{CV}$ (calculated as $y_{CV} = f(x_{CV}) = f(Temp_{CV}(Q, y_C, A'))$) selects the option for $y_C$ in $x_{CV}$. Similarly, perceptual task consistency quantifies the percentage of cases where $y_{PV}$ (calculated as $y_{PV} = f(x_{PV}) = f(Temp_{PV}(Q, y_P, Box'))$) selects the option for $y_P$ in $x_{PV}$.

The experimental results, as illustrated in Table 3 and Table 4, demonstrate that our multimodal knowledge consistency fine-tuning method substantially improves C&P consistency across all six datasets. Specifically, Qwen-VL-Chat exhibits a 34.83% increase in C&P consistency, while InternVL2-2b and InternVL2-8b show improvements of 37.85% and 43.19%, respectively. These results indicate that our method effectively reduces C&P knowledge conflicts. The comparison between Qwen-VL-Chat and the InternVL2 models highlights the general applicability of our approach across different MLLM architectures. The results reveal that models with a larger number of parameters, such as InternVL2-8b, achieve better C&P consistency after fine-tuning.

Table 3: Performance comparison between the original MLLM and the MLLM after multimodal knowledge consistency fine-tuning (Ours). Only micro-average results are presented, with detailed results for each dataset in Table 4. "C" stands for Cognitive Task Consistency, "P" stands for Perceptual Task Consistency, and "C&P" stands for C&P Consistency. All values are reported as percentages (%), with bolded numbers indicating superior performance.

| | Average | | |
| --- | --- | --- | --- |
| | C | P | C&P |
| Qwen-VL-Chat | 56.23 | 52.35 | 19.41 |
| Qwen-VL-Chat (Ours) | **98.59** | **97.51** | **54.24** |
| InternVL2-2b | 54.07 | 54.30 | 12.09 |
| InternVL2-2b (Ours) | **99.19** | **95.95** | **49.94** |
| InternVL2-8b | 67.43 | 75.40 | 16.87 |
| InternVL2-8b (Ours) | **99.76** | **96.75** | **60.03** |

Table 4: Performance comparison between the original MLLM and the MLLM after multimodal knowledge consistency fine-tuning (ours) across all datasets. Average results are presented in Table 3. All values are reported as percentages (%), with bolded numbers indicating superior performance.

| | DocVQA | | | DeepForm | | | KLC | | |
| --- | --- | --- | --- | --- | --- | --- | --- | --- | --- |
| | C | P | C&P | C | P | C&P | C | P | C&P |
| Qwen-VL-Chat | 56.53 | 51.66 | 20.82 | 52.11 | 57.79 | 5.240 | 50.99 | 49.01 | 37.87 |
| Qwen-VL-Chat (Ours) | **98.90** | **97.36** | **56.05** | **99.27** | **95.20** | **37.12** | **99.51** | **98.76** | **70.55** |
| InternVL2-2b | 53.69 | 47.65 | 13.92 | 45.71 | 45.71 | 1.456 | 59.98 | 62.13 | 18.48 |
| InternVL2-2b (Ours) | **99.52** | **95.07** | **41.07** | **100.0** | **96.22** | **44.54** | **99.92** | **97.11** | **76.40** |
| InternVL2-8b | 47.50 | 79.48 | 20.47 | 52.84 | 79.04 | 3.202 | 85.48 | 81.35 | 30.53 |
| InternVL2-8b (Ours) | **99.90** | **95.39** | **52.21** | **100.0** | **96.94** | **44.69** | **100.0** | **98.52** | **81.68** |
| | FUNSD | | | ChartQA | | | WTQ | | |
| | C | P | C&P | C | P | C&P | C | P | C&P |
| Qwen-VL-Chat | 55.46 | 53.27 | 7.264 | 58.37 | 51.67 | 21.64 | 56.83 | 54.68 | 8.672 |
| Qwen-VL-Chat (Ours) | **95.93** | **97.58** | **45.04** | **98.88** | **98.92** | **72.68** | **97.97** | **96.88** | **32.59** |
| InternVL2-2b | 46.47 | 59.32 | 7.506 | 48.84 | 46.89 | 9.107 | 58.00 | 66.46 | 10.30 |
| InternVL2-2b (Ours) | **98.93** | **95.40** | **26.63** | **99.04** | **98.38** | **78.81** | **98.57** | **95.60** | **39.97** |
| InternVL2-8b | 91.44 | 81.60 | 11.14 | 50.28 | 47.25 | 9.558 | 95.76 | 77.03 | 9.214 |
| InternVL2-8b (Ours) | **99.79** | **94.43** | **37.29** | **99.56** | **99.46** | **83.86** | **99.59** | **97.56** | **59.35** |

## 5.3 ABLATION STUDY

To further evaluate the effectiveness of multimodal knowledge consistency fine-tuning, we conducted a series of ablation experiments using Qwen-VL-Chat, as shown in Table 5. Each experiment, with different fine-tuning tasks, is trained according to the settings outlined in Section 5.1. The results validate our hypothesis that both task-specific consistency and the integration of cognitive and perceptual knowledge are crucial for enhancing C&P consistency. For instance, in terms of average results, the perception consistency task improves by 14.79%, the cognition consistency task improves by 0.44%, and the C&P connector improves by 1.06%. It is observed that the perception consistency task demonstrates the largest gain, likely due to the limited perception capabilities of open-source MLLMs, as discussed in Section 5.4.

## 5.4 THE PERFORMANCE OF CONGITIVE AND PERCEPTUAL TASKS

Improving C&P consistency does not necessarily correlate with enhanced performance in cognitive and perceptual tasks, as an MLLM can exhibit consistency even if both cognitive and perceptual outputs are incorrect. Therefore, Table 6 presents the MLLM's performance on cognitive and perceptual

Table 5: Ablation study based on Qwen-VL-Chat. C&P consistency is reported as percentages (%). The best results are in bold. "Cog.", "Per.", and "Conn." stand for Cognition Consistency task, Perception Consistency task, and C&P Connector task, respectively, as detailed in Section 4.

| # | Per. | Cog. | Conn. | Doc VQA | Deep Form | KLC | FUNSD | Chart QA | WTQ | Average |
|---|------|------|-------|---------|-----------|-----|-------|----------|-----|---------|
| 1 |      | ✓    | ✓     | 36.39   | 16.59     | 62.05 | 26.15 | 63.57  | 25.54 | 39.45 |
| 2 | ✓    |      | ✓     | 54.55   | **39.74** | **72.36** | **45.04** | 71.96 | 31.84 | 53.80 |
| 3 | ✓    | ✓    |       | 54.52   | 35.37     | 68.98 | 43.58 | 72.32  | **33.13** | 53.18 |
| 4 | ✓    | ✓    | ✓     | **56.05** | 37.12   | 70.55 | **45.04** | **72.68** | 32.59 | **54.24** |

Table 6: The performance of cognitive and perceptual tasks. "C.T." and "P.T." stand for cognitive task (VQA) and perceptual task (OCR), respectively. The metrics are detailed in Section 5.4, and all values are reported as percentages (%), with bolded numbers indicating superior performance.

| | Doc VQA | | Deep Form | | KLC | | FUNSD | | Chart QA | | WTQ | |
|---|------|------|------|------|------|------|------|------|------|------|------|------|
| | C.T. | P.T. | C.T. | P.T. | C.T. | P.T. | C.T. | P.T. | C.T. | P.T. | C.T. | P.T. |
| Qwen-VL-Chat | 62.5 | 22.7 | 4.22 | 9.07 | 47.1 | 48.6 | 47.5 | 11.0 | 63.5 | 27.2 | 22.4 | 11.5 |
| Qwen-VL-Chat (Ours) | **63.5** | **74.2** | **34.4** | **66.7** | **63.0** | **89.2** | **50.3** | **62.0** | **63.7** | **96.6** | **23.7** | **76.5** |
| InternVL2-2b | **87.0** | 13.9 | 35.1 | 3.30 | 68.8 | 25.1 | **74.0** | 9.81 | **76.3** | 10.4 | 35.1 | 11.4 |
| InternVL2-2b (Ours) | 84.6 | **46.4** | **88.8** | **56.7** | **83.9** | **88.4** | 73.8 | **27.8** | 75.3 | **92.6** | **36.7** | **70.7** |
| InternVL2-8b | **91.7** | 20.6 | 38.4 | 5.14 | 72.9 | 37.7 | 75.8 | 12.1 | **83.2** | 9.89 | 49.2 | 11.3 |
| InternVL2-8b (Ours) | 89.3 | **57.0** | **90.5** | **58.6** | **86.5** | **92.6** | **76.3** | **39.7** | 81.1 | **95.6** | **51.2** | **84.0** |

tasks. For the cognitive task, following previous works (Borchmann et al., 2021; Lee et al., 2023; Luo et al., 2024), we evaluate DocVQA and FUNSD using ANLS (Biten et al., 2019), DeepForm and KLC using the F1 score, and ChartQA using relaxed accuracy (Methani et al., 2020). WTQ is evaluated based on accuracy. For the perceptual task, all datasets are evaluated using ANLS.

The results in Table 6 demonstrate that multimodal knowledge consistency fine-tuning does not degrade the performance of the MLLM in most scenarios. Specifically, for Qwen-VL-Chat, improvements are observed in both cognitive and perceptual tasks across all datasets after fine-tuning. Similarly, InternVL2-2B and InternVL2-8B show enhanced performance on most datasets, with only minor declines in cognitive tasks on a few datasets. We attribute this improvement to our fine-tuning approach, which integrates perceptual and cognitive knowledge within the MLLM. Additionally, it is observed that before fine-tuning, performance on perceptual tasks is significantly weaker than on cognitive tasks, further confirming that cognition is not consistent with perception in current open-source MLLMs.

## 5.5 Case Study

Figure 4 presents two examples generated by Qwen-VL-Chat. In both cases, the original C&P conflicts are resolved after fine-tuning, highlighting the effectiveness of multimodal knowledge consistency fine-tuning. Notably, in Figure 4 (a), both cognitive and perceptual responses remain incorrect after fine-tuning, which explains the observed performance decline in some datasets (Table 6). However, cases like Figure 4 (a) are not general, and considering the substantial improvement in C&P consistency after fine-tuning, such "trade-offs" are considered acceptable.

## 6 Related Work

### 6.1 Multimodal Large Language Models

With the advancement of large language models (LLMs; (Brown et al., 2020; Touvron et al., 2023)), researchers are investigating the integration of vision and other modalities into LLMs (gpt, 2023; Team et al., 2023; Liu et al., 2023b; Ye et al., 2024; Bai et al., 2023; Chen et al., 2024). These mul-

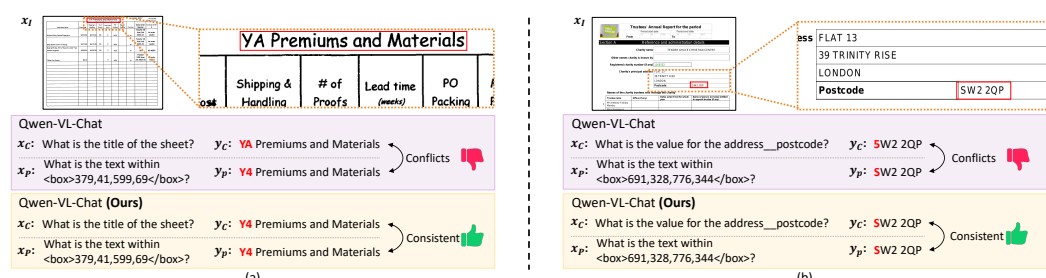

Figure 4: Two cases illustrate the effectiveness of our method.

timodal large language models (MLLMs) possess the capability to perceive visual content, perform visual reasoning, and engage in multimodal dialogues with humans. Following this, models such as the LLaVA series (Liu et al., 2023b), and MiniGPT-4 (Zhu et al., 2024) have introduced visual instruction tuning to enhance the instruction-following abilities of vision-language models. Concurrently, models like InternVL, Qwen-VL (Bai et al., 2023; Chen et al., 2024) have augmented MLLMs with advanced visual capabilities, thereby improving performance on vision-language tasks. These developments highlight significant advancements in MLLMs.

## 6.2 MLLMs for Document Understanding

Document understanding (Cui et al., 2021; Xu et al., 2020; 2021; Huang et al., 2022; Gu et al., 2022; Luo et al., 2023; 2024; Wang et al., 2023) is a rapidly growing research area driven by increasing industrial demand. Its main objective is to comprehend complex typeset images that contain rich textual information, such as scanned document pages (Mathew et al., 2021; Svetlichnaya, 2020; Stanisławek et al., 2021), charts (Masry et al., 2022; Kafle et al., 2018; Methani et al., 2020), tables (Pasupat & Liang, 2015; Chen et al., 2019), and other formats (Tanaka et al., 2021; Mathew et al., 2022). As a multimodal task, document understanding involves automated processes for understanding, classifying, and extracting information, requiring models to possess both perceptual and cognitive capabilities (Cui et al., 2021). Recent studies (Chen et al., 2024; Hong et al., 2024; Dong et al., 2024) for general MLLMs improve the encoding resolution of document images, significantly boosting performance in document understanding tasks. Several MLLMs are developed to focus on addressing document understanding problems. mPLUG-DocOwl (Ye et al., 2023a; Hu et al., 2024) and UReader (Ye et al., 2023b) unify tasks across five types of document images using a sequence-to-sequence format, and achieve good performance in document understanding.

## 6.3 Knowledge Conflicts in LLMs

LLMs are distinguished for encapsulating an extensive repository of world knowledge, known as the memory. Simultaneously, LLMs continue to engage with external contextual knowledge post-deployment (Pan et al., 2023). The discrepancies between the contexts and the model's memory knowledge, i.e. context-memory conflicts, are being intensively studied recently (Xie et al., 2023; Jin et al., 2024). Another notable challenge arises with intra-memory conflict—a condition where LLMs exhibit unpredictable behaviors to inputs that are semantically equivalent but syntactically distinct (Chang & Bergen, 2023; Chen et al., 2023; Raj et al., 2023; Rabinovich et al., 2023; Bartsch et al., 2023). This variance can be attributed to the conflicting knowledge embedded within the LLM's memory, which stem from the inconsistencies present in the complex and diverse pre-training data sets. However, current research on knowledge conflicts focuses only on text, leaving the issue of multimodal knowledge conflicts in MLLMs unaddressed.

## 6.4 Hallucination issues in MLLMs

MLLMs provide powerful tools for content generation across a wide range of tasks. However, they are susceptible to hallucinations (Bang et al., 2023; Zhang et al., 2023c; Guan et al., 2024; Li et al., 2023), where the generated outputs contain information not present in the visual input. These hallucinations typically arise when the models overly rely on the strong priors of their language modules,

neglecting visual sensibility (Guan et al., 2024). Such conflicts between MLLMs' language and visual perception raise concerns about their reliability and limit their applications (Ji et al., 2023; Kaddour et al., 2023). Current research primarily focuses on detecting and evaluating hallucinations (Li et al., 2023; Zhang et al., 2023b;c), as well as methods to reduce them (Liu et al., 2024; Wang et al., 2024). To mitigate hallucinations, efforts have been directed toward enhancing data collection and training procedures. For instance, LRV-Instruction (Liu et al., 2024) creates balanced positive and negative instructions to finetune MLLMs, while VIGC (Wang et al., 2024) employs an iterative process to generate concise answers and combine them. These approaches equip the model with more accurate perception capability. Nevertheless, research on how MLLMs integrate perception and cognition knowledge, which is also vital for interpreting and debugging these models, has not progressed at the same pace.

## 7    CONCLUSION

In this paper, we identify that current MLLMs often face conflicts between perception and cognition, referred to as *Cognition and Perception (C&P) knowledge conflicts*. The severity of these conflicts is systematically assessed across six document understanding datasets, revealing that even leading MLLMs still struggle with these multimodal knowledge conflicts. To address this problem, a novel method called *Multimodal Knowledge Consistency Fine-tuning* is introduced. Comprehensive experiments demonstrate the effectiveness of our method in reducing C&P knowledge conflicts. Additionally, in most scenarios, our method improves the performance of MLLMs in both cognitive and perceptual tasks. One limitation of our work is its focus solely on document understanding. In the future, we will expand our research beyond document understanding to examine C&P knowledge conflicts in more general multimodal areas, such as scene understanding and visual reasoning.

## REPRODUCIBILITY STATEMENT

We fully recognize the importance of reproducibility and make significant efforts to ensure it. All the datasets we use are publicly available (Section 2.2), with the data construction process described in detail in Sections 2.3 and 3. For the models, Section 3 provides links to the APIs and weights we use. In terms of fine-tuning, Section 5.1 outlines the implementation details, and the fine-tuning code directly follows official repositories. We hope these efforts contribute to the reproducibility of this work.

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
