# OpenReview forum: "Is Cognition consistent with Perception? Assessing and Mitigating Multimodal Knowledge Conflicts in Document Understanding"
_ICLR.cc/2025/Conference — ICLR 2025 Conference Withdrawn Submission_

### Official Review · Reviewer_fQT2 · 2024-11-03

**Soundness:** 2
**Presentation:** 2
**Contribution:** 2
**Rating:** 3
**Confidence:** 4

**Summary:**

This paper introduces the concept of Cognition and Perception Knowledge Conflicts to measure the discrepancy between perceptual and cognitive abilities in multimodal large language models (MLLMs). Through experiments focused on document understanding, it demonstrates a significant gap between these two abilities. Additionally, the paper proposes a tuning method, Multimodal Knowledge Consistency Fine-tuning, which effectively reduces this gap, mitigating the conflict between cognitive and perceptual knowledge.

**Strengths:**

- This paper introduces a new concept, Cognition and Perception Knowledge Conflicts, to investigate the behavioral discrepancy between the cognitive and perceptual abilities of MLLMs.

- This paper proposes a tuning method to mitigate the conflict by constructing a new dataset for training.

**Weaknesses:**

- The concept of Cognition and Perception knowledge conflicts is not entirely convincing:
  - The task used to measure the model's cognitive abilities does not seem to require much cognitive processing.
  - The tasks used in these experiments are limited in scope.

- It is unclear whether the low C&P consistencies arise from C&P knowledge conflicts or simply from poor model performance:
  - The varying consistency scores across different datasets in Table 2 suggest that the conflict may not be an inherent property of the model itself but rather linked to its performance on specific tasks.
  - Empirical results in Table 4 show that improvements in C&P performance appear related to increases in cognitive and perceptual abilities, supporting the idea that C&P knowledge conflicts might stem from the models' poor performance.

**Questions:**

Line 071 : What exactly does 'the intuitive notion that cognition is consistent with perception' mean?

---

> ### Author Response · Authors · 2024-11-21
>
> Thank you for dedicating your time to reviewing our paper and offering insightful feedback. We appreciate your recognition of the significance of C&P knowledge conflicts and the effectiveness of our tuning method. We have carefully considered your concerns and addressed them in the following response.
>
>
>
> > The task used to measure the model's cognitive abilities does not seem to require much cognitive processing.
> >
>
> Thank you for your question. We wish to clarify that although all VQA problems in the evaluation dataset (Section 2.3) require the text on the image as the answer, this does not imply that they can be solved by simple information extraction.  In fact, **a considerable number of VQA problems demand reasoning**, such as comparison and counting. We provide an example that requires reasoning [here](https://duguang-netdisk.oss-cn-hangzhou.aliyuncs.com/hangdi/tmp/image2.png). We will include relevant explanations in the revision to clarify this point.
>
>
>
> > The tasks used in these experiments are limited in scope.
> >
>
> We appreciate your question regarding the scope of the experiment.  Document understanding is a rapidly expanding research area with growing demand in industry, and our experiment provides comprehensive coverage of document understanding by including four tasks (Section 2.2). Additionally, to the best of our knowledge, we are **the first to identify and introduce the concept of C&P knowledge conflicts**, which affect even leading MLLMs. We believe document understanding is an ideal domain to demonstrate C&P knowledge conflicts. Moreover, our evaluation and fine-tuning methods **can be readily applied to other multimodal tasks,** such as using object detection as the perceptual task and general VQA as the cognitive task in scene understanding. As mentioned in Section 7, we intend to investigate this direction in future work.
>
>
>
> > It is unclear whether the low C&P consistencies arise from C&P knowledge conflicts or simply from poor model performance:
> >
> > + The varying consistency scores across different datasets in Table 2 suggest that the conflict may not be an inherent property of the model itself but rather linked to its performance on specific tasks.
> > + Empirical results in Table 4 show that improvements in C&P performance appear related to increases in cognitive and perceptual abilities, supporting the idea that C&P knowledge conflicts might stem from the models' poor performance.
> >
>
>
>
> Thank you for raising this question. The performance of closed-source MLLMs on OCR and VQA tasks in Table 2, evaluated using ANLS, is as follows:
>
> |  |  | DocVQA | DeepForm | KLC | FUNSD | ChartQA | WTQ | **Average** |
> | --- | --- | --- | --- | --- | --- | --- | --- | --- |
> | GPT-4o | VQA | 92.50 | 53.26 | 88.60 | 82.60 | 81.68 | 69.65 | **82.41** |
> | GPT-4o | OCR | 80.21 | 42.47 | 92.72 | 84.22 | 95.96 | 78.51 | **81.27** |
> | Qwen-VL-Max | VQA | 95.33 | 73.48 | 92.61 | 81.41 | 88.20 | 74.77 | **87.25** |
> | Qwen-VL-Max | OCR | 82.46 | 57.52 | 90.99 | 85.32 | 91.79 | 88.57 | **84.78** |
>
>
> It can be observed that closed-source MLLMs perform remarkably well on both VQA and OCR tasks. However, despite their strong performance, they still exhibit C&P knowledge conflicts, suggesting **a relationship between these conflicts and the models' inherent properties**.
>
> Furthermore, we acknowledge that the current definition of C&P knowledge conflicts may related to the poor performance of MLLMs, especially open-source ones, an effect we cannot entirely isolate during experiments. Notably, in the Table 2 experiment, only 34.4% of inconsistent cases for Qwen-VL-Chat involve incorrect results in both OCR and VQA tasks.
>
> Additionally, as defined in Section 2.1, all VQA problems in the evaluation dataset (Section 2.3) require the text on the image as the answer. This means that if the model correctly answers a VQA question, it definitely recognizes the corresponding text. Therefore, **cases where the model answers correctly on VQA tasks but incorrectly on OCR tasks are not due to poor model performance**. In the Qwen-VL-Chat experiment shown in Table 2, up to 53.7% of cases fall into this category, indicating that **the inherent properties of MLLMs are the main cause of C&P inconsistencies**. We will include a detailed analysis in the revision.

---

> > ### Author Response · Authors · 2024-11-21
> >
> > > Line 071 : What exactly does 'the intuitive notion that cognition is consistent with perception' mean?
> > >
> >
> > Human cognition is intuitively built upon perception. Similar to how humans process information, as mentioned in line 59, document understanding requires models to accurately perceive visual content (perception) and generate coherent responses (cognition) based on that perception. In other words, for a model to answer VQA, it must "see" the answer text within the image, meaning the OCR results should be consistent with the VQA results. However, C&P inconsistency challenges this intuitive concept. Additional explanations will be included in the revision for better clarity.
> >
> >
> >
> >
> >
> > Finally, we would like to express our gratitude once again for your review and look forward to further discussions with you.

---

> > > ### Comment · Reviewer_fQT2 · 2024-11-23
> > >
> > > Thank you for the response. However, after reviewing the authors' response, I remain unconvinced about the concept of Cognition and Perception Knowledge Conflicts. Additionally, the influence of model performance on these conflicts has not been clearly clarified. Therefore, I will maintain my original score.

---

### Official Review · Reviewer_iD1S · 2024-11-03

**Soundness:** 2
**Presentation:** 3
**Contribution:** 3
**Rating:** 5
**Confidence:** 3

**Summary:**

This submission discusses the concept of Cognition and Perception knowledge conflicts in MLLMs and proposes a method to mitigate these conflicts in document understanding tasks. ​ The proposed method includes three fine-tuning tasks aimed at improving consistency between cognitive and perceptual knowledge. ​ Experimental results demonstrate that this approach significantly enhances C&P consistency and overall performance in both cognitive and perceptual tasks.

**Strengths:**

There are several noticeable strengths in this paper:
* This paper sets eye on a new research area and presents a well-justified motivation for examining the conflicts between cognition and perception knowledge of MLLMs.
* This submission conducts extensive experiments to examine current open-source and closed-source MLLMs performance on the cognition and perception knowledge conflict problems, and to verify the effectiveness of the proposed finetuning method.
* The manuscript is commendable for its clarity and structured writing style, which greatly facilitates reader comprehension.
* Additionally, the inclusion of clear and illustrative figures and tables is a notable strength, as it significantly aids in conveying the main claims of the paper to the audience.

**Weaknesses:**

My main criticism for this submission is on its experimental design:

* **Limited Reproducibility**: Although the submission provides links to the model and weights, details about the hyperparameters used during inference are missing (e.g., values for top_p, top_k, temperature, beam search, etc). This lack of transparency makes it difficult to ensure fair comparisons across models and affects reproducibility.
* **Lack of Repeated Experiments**: The authors did not report the mean and standard deviation of the experimental results, nor did they specify any settings for repeated runs, suggesting that results may have been obtained from a single inference run. However, LLMs and MLLMs often yield varying outputs across runs due to their inherent generation variability. How do the authors account for this potential source of instability when attributing results solely to conflicts in cognitive or perceptual knowledge, rather than to the natural output variability of the models?

**Questions:**

Please refer to the Weaknesses above.

---

> ### Author Response · Authors · 2024-11-21
>
> Thank you for taking the time to review our paper and providing valuable feedback. We sincerely appreciate your positive remarks regarding the motivation behind investigating C&P knowledge conflicts and the effectiveness of our fine-tuning method. We have carefully reviewed your concerns and addressed them in detail below.
>
>
>
> > Limited Reproducibility: Although the submission provides links to the model and weights, details about the hyperparameters used during inference are missing (e.g., values for top_p, top_k, temperature, beam search, etc). This lack of transparency makes it difficult to ensure fair comparisons across models and affects reproducibility.
> >
>
>
>
> Thank you for your valuable suggestions. The hyperparameters used during inference for each MLLM are set to their default values, which are available on their official websites or repositories. We apologize for the omission and will include these details in the revision.
>
>
>
> > Lack of Repeated Experiments: The authors did not report the mean and standard deviation of the experimental results, nor did they specify any settings for repeated runs, suggesting that results may have been obtained from a single inference run. However, LLMs and MLLMs often yield varying outputs across runs due to their inherent generation variability. How do the authors account for this potential source of instability when attributing results solely to conflicts in cognitive or perceptual knowledge, rather than to the natural output variability of the models?
> >
>
> We acknowledge the inherent generation variability in MLLMs,  leading to potential randomness in their responses. For open-source MLLMs, we adhere to their default settings and **disable all hyperparameters that could introduce randomness during inference**. For instance, we set `do_sample` to `False` and `temperature` to `0`. Therefore, repeated experiments on open-source MLLMs are unnecessary. These details will be added in the revision to clarify this point.
>
>
>
> For closed-source MLLMs, it is not possible to completely eliminate response randomness due to official restrictions. Following your suggestion, we repeat the experiments three times using GPT-4o and report the mean and standard deviation for C&P consistency.
>
> |  | DocVQA | DeepForm | KLC | FUNSD | ChartQA | WTQ | **Average** |
> | --- | --- | --- | --- | --- | --- | --- | --- |
> | GPT-4o | 75.59 ± 0.48 | 16.71 ± 0.44 | 79.84 ± 0.37 | 65.61 ± 0.39 | 63.86 ± 0.08 | 55.83 ± 0.45 | **65.51 ± 0.14** |
>
>
> The observed standard deviations are small, demonstrating that **the conclusion regarding leading MLLMs like GPT-4o exhibiting C&P knowledge conflicts remains robust**. These experimental results will be included in the revision.
>
>
>
> Additionally, we note that during the rebuttal period, GPT-4o's performance differs from the results of our September experiments as reported in the paper, possibly due to updates made to GPT-4o during this time. We emphasize that these differences do not affect our experimental conclusions.
>
>
>
> Once again, we would like to thank you for your constructive feedback, and we look forward to your further response.

---

### Official Review · Reviewer_HHQe · 2024-11-03

**Soundness:** 3
**Presentation:** 3
**Contribution:** 2
**Rating:** 5
**Confidence:** 4

**Summary:**

Document understanding, as a multimodal task, requires models to accurately perceive visual content (perception) and then generate coherent responses (cognition) based on this perception. However, current multimodal large language models (MLLMs) often encounter conflicts between perception and cognition. For example, a model may use its OCR capabilities (perception) to recognize text within an image as "A" but then respond to a related question by providing a similar yet incorrect word, "B." In this paper, the authors define these intrinsic conflicts between cognitive knowledge and perceptual knowledge within MLLMs as *Cognition and Perception (C&P) knowledge conflicts*, leading to inconsistent responses involving cognition and perception. They conduct a systematic evaluation of current MLLMs to assess the prevalence of these conflicts in document understanding, demonstrating that such conflicts are indeed widespread. To address this issue, the authors propose a fine-tuning method called *Multimodal Knowledge Consistency Fine-tuning*, which aims to mitigate these conflicts.

**Strengths:**

1. **Clear Experimental Evidence:** The study provides a clear statistical presentation of answer inconsistencies in MLLM responses when different question formats are used, effectively highlighting the inconsistency issue in MLLMs.

2. **Definition of C&P Knowledge Conflicts:** The paper attempts to systematically define these inconsistencies as *Cognition and Perception (C&P) knowledge conflicts*, offering a detailed definition of the term.

3. **Systematic Evaluation of Existing Models:** It conducts a comprehensive assessment of existing MLLMs to evaluate the prevalence and nature of C&P knowledge conflicts, offering valuable insights into current model limitations.

4. **Proposed Fine-Tuning Method:** The authors propose a fine-tuning method incorporating three specific training tasks, aimed at enhancing consistency between cognition and perception in MLLMs.

**Weaknesses:**

1. **Lack of Robust Explanation for C&P Conflicts:** The attribution of answer inconsistencies across different questioning formats solely to Cognition and Perception (C&P) knowledge conflicts lacks a thorough explanation. The authors could strengthen this claim by including experiments that control for potential randomness in MLLM responses. For example, they could ask the same question multiple times with identical phrasing to observe whether inconsistencies persist under uniform questioning formats, which would help determine if the issue is rooted in C&P conflicts or random model behavior.

2. **Insufficient Consideration of OCR Limitations in Defining C&P Conflicts:** Given that high OCR accuracy is critical for intensive document understanding tasks, the definition of C&P knowledge conflicts should more carefully account for OCR limitations. The authors might include experiments to isolate and assess OCR capabilities, such as by using extracted OCR text as an auxiliary input (similar to the approach in TextVQA datasets) or by implementing other techniques to reduce OCR-related inconsistencies. This would help clarify whether observed C&P conflicts stem from limitations in OCR or from other aspects of the model's perception and cognition alignment.

3. **Limited Novelty in Fine-Tuning Approach:** While the proposed fine-tuning method appears effective in reducing inconsistencies, it essentially involves fine-tuning a general-purpose multimodal LLM on a document-understanding dataset, resulting in a specialized model for this particular task. Although this is a practical solution, it represents a relatively common approach in MLLM research and may lack innovation. Additionally, this specialization may come at the cost of the model's broader capabilities as a general-purpose MLLM. To enhance the study's impact, the authors could address these limitations by presenting results on more general benchmarks, which would demonstrate that their fine-tuning method improves consistency in document understanding without sacrificing the model's performance on other tasks, thereby reinforcing its relevance for broader MLLM applications.

**Questions:**

See weakness.

---

> ### Author Response · Authors · 2024-11-21
>
> Thank you for dedicating your time to reviewing our paper and offering insightful feedback. We are grateful for your recognition of the value of C&P knowledge conflicts and our fine-tuning method. About the concerns you raised, we have carefully considered them and would like to address them in the following response.
>
>
>
> > Lack of Robust Explanation for C&P Conflicts: The attribution of answer inconsistencies across different questioning formats solely to Cognition and Perception (C&P) knowledge conflicts lacks a thorough explanation. The authors could strengthen this claim by including experiments that control for potential randomness in MLLM responses. For example, they could ask the same question multiple times with identical phrasing to observe whether inconsistencies persist under uniform questioning formats, which would help determine if the issue is rooted in C&P conflicts or random model behavior.
> >
>
>
>
> We appreciate your concern regarding the random behavior of MLLMs. For open-source MLLMs, we adhere to their default settings and **disable all hyperparameters that could introduce randomness during inference**. For instance, we set `do_sample` to `False` and `temperature` to `0`. Therefore, we can confidently conclude that **the inconsistency in open-source MLLMs is not caused by response randomness**.These details will be added in the revision to clarify this point.
>
>
>
> For closed-source MLLMs, it is not possible to completely eliminate response randomness due to official restrictions. Following your suggestion, we repeat the experiments three times using GPT-4o and report the mean and standard deviation for C&P consistency.
>
> |  | DocVQA | DeepForm | KLC | FUNSD | ChartQA | WTQ | **Average** |
> | --- | --- | --- | --- | --- | --- | --- | --- |
> | GPT-4o | 75.59 ± 0.48 | 16.71 ± 0.44 | 79.84 ± 0.37 | 65.61 ± 0.39 | 63.86 ± 0.08 | 55.83 ± 0.45 | **65.51 ± 0.14** |
>
>
> The observed standard deviations are small, demonstrating that **the conclusion regarding leading MLLMs like GPT-4o exhibiting C&P knowledge conflicts remains robust**. These experimental results will be included in the revision.
>
>
>
> Additionally, we note that during the rebuttal period, GPT-4o's performance differs from the results of our September experiments as reported in the paper, possibly due to updates made to GPT-4o during this time. We emphasize that these differences do not affect our experimental conclusions.

---

> > ### Author Response · Authors · 2024-11-21
> >
> > > Insufficient Consideration of OCR Limitations in Defining C&P Conflicts: Given that high OCR accuracy is critical for intensive document understanding tasks, the definition of C&P knowledge conflicts should more carefully account for OCR limitations. The authors might include experiments to isolate and assess OCR capabilities, such as by using extracted OCR text as an auxiliary input (similar to the approach in TextVQA datasets) or by implementing other techniques to reduce OCR-related inconsistencies. This would help clarify whether observed C&P conflicts stem from limitations in OCR or from other aspects of the model's perception and cognition alignment.
> > >
> >
> >
> >
> > Thank you for raising this question. The performance of closed-source MLLMs on the OCR task (evaluated using ANLS) in Table 2 is as follows:
> >
> > |  | DocVQA | DeepForm | KLC | FUNSD | ChartQA | WTQ | **Average** |
> > | --- | --- | --- | --- | --- | --- | --- | --- |
> > | GPT-4o | 80.21 | 42.47 | 92.72 | 84.22 | 95.96 | 78.51 | **81.27** |
> > | Qwen-VL-Max | 82.46 | 57.52 | 90.99 | 85.32 | 91.79 | 88.57 | **84.78** |
> >
> >
> > We observe that closed-source MLLMs perform remarkably well on OCR tasks. However, **despite their strong OCR capabilities, they still exhibit C&P knowledge conflicts**, highlighting the importance of addressing this issue.
> >
> > **Please note that our paper investigates consistency issues**.  Even if the model's OCR misrecognizes text, but when performing VQA based on the OCR result it provides the same incorrect answer, we still consider this case C&P consistent.  This parallels human behavior: when a person misreads text, their answer based on that error is naturally incorrect. However, this does not negate that human cognition relies on perception.
> >
> > Regarding the use of extracted OCR text as an auxiliary input, this introduces a new type of conflict:  **conflicts between the MLLM's own OCR perception and the input external OCR**, similar to the context-memory conflicts in LLMs [1]. In contrast, **C&P knowledge conflicts resemble intra-memory conflicts in LLMs** [1].
> >
> > To investigate this new type of conflict, we conduct a simple experiment with 148 samples from DocVQA using GPT-4o. We find that the model's performance is unstable when both the image and the official DocVQA OCR text are provided. In 19.5% of cases, the model relies on external OCR to generate responses, whereas in 65.8% of cases, it does not, leading us to hypothesize that the latter responses are derived from the model’s internal perception.
> >
> > **In this work, we focus on C&P knowledge conflicts**, and to avoid introducing new conflicts, we decide not to use external OCR inputs in our experiments while planning to explore these conflicts in future work.
> >
> >
> >
> > > Limited Novelty in Fine-Tuning Approach: While the proposed fine-tuning method appears effective in reducing inconsistencies, it essentially involves fine-tuning a general-purpose multimodal LLM on a document-understanding dataset, resulting in a specialized model for this particular task. Although this is a practical solution, it represents a relatively common approach in MLLM research and may lack innovation. Additionally, this specialization may come at the cost of the model's broader capabilities as a general-purpose MLLM. To enhance the study's impact, the authors could address these limitations by presenting results on more general benchmarks, which would demonstrate that their fine-tuning method improves consistency in document understanding without sacrificing the model's performance on other tasks, thereby reinforcing its relevance for broader MLLM applications.
> > >
> >
> > We appreciate your question. To the best of our knowledge, we are **the first to identify and introduce the concept of C&P knowledge conflicts**, which affect even leading MLLMs. We believe document understanding is an ideal domain to demonstrate C&P knowledge conflicts.
> >
> > Furthermore, unlike related works (e.g., hallucination) that focus solely on perception, **our work highlights the consistency between perception and cognition**. Our proposed fine-tuning method effectively establishes a connection between the two.
> >
> > Additionally, we acknowledge that our fine-tuning may reduce the broader capabilities of MLLMs. However, our methods **can readily apply to other multimodal tasks**, such as using object detection for perception and general VQA for cognitive tasks in scene understanding. Due to resource limitations, such as computational resources, we currently conduct experiments only in document understanding and plan to investigate this direction further in future work, as mentioned in Section 7.
> >
> >
> >
> > Again, thank you for your time and effort in providing a comprehensive review, and we look forward to your further response.
> >
> >
> >
> > ### Reference
> > [1] Xu, Rongwu, et al. "Knowledge conflicts for llms: A survey." arXiv preprint arXiv:2403.08319 (2024).

---

> > > ### Comment · Reviewer_HHQe · 2024-12-02
> > >
> > > I greatly appreciate your patience and the detailed response. After carefully reviewing the rebuttal, I find that while some of my concerns have been addressed, my primary concern regarding **C&P knowledge conflicts** remains unresolved. Consequently, I will maintain my current score.
> > >
> > > First, I would like to thank the authors for clarifying the experimental settings related to randomness. This has alleviated my concerns about the potential role of randomness in causing C&P knowledge conflicts.
> > >
> > > Second, OCR plays a pivotal role in document understanding tasks and must be carefully considered to eliminate its potential influence on C&P knowledge conflicts. Introducing a new concept like C&P knowledge conflicts without fully addressing the impact of OCR capabilities risks conflating pre-existing phenomena with the proposed framework. While the idea of C&P knowledge conflicts is intriguing, its value depends on demonstrating that it represents a genuinely novel insight rather than a reinterpretation of existing observations.
> > >
> > > Finally, concerns regarding generalization persist.

---

### Official Review · Reviewer_kooV · 2024-11-05

**Soundness:** 2
**Presentation:** 2
**Contribution:** 2
**Rating:** 5
**Confidence:** 4

**Summary:**

This paper explores the capability of large multimodal models in understanding document images. The authors highlight that current multimodal large models often provide conflicting extraction results in "perception" tasks (i.e., OCR) and "cognition" tasks (i.e., VQA based on document images). To address this issue, the authors introduce a metric to compute cognition-perception (C&P) consistency and construct a multi-task fine-tuning dataset based on OCR annotations. Experiments demonstrate that the proposed fine-tuning method significantly improves C&P consistency across three models with sizes ranging from 2B to 8B parameters.

**Strengths:**

1. This work highlights a noteworthy issue: an (M)LLM's factual understanding of identical content can vary depending on contextual differences.
2. The joint OCR-VQA dataset proposed in this paper may benefit future research in related topics.

**Weaknesses:**

1. Insufficient Experimental Results on C&P Conflict Causes and Impact: The paper does not provide extensive experimental results on the root causes of C&P conflicts or a detailed analysis of their impact on downstream tasks. While it demonstrates the existence of these conflicts and the effectiveness of the proposed method in reducing them, a deeper investigation into why these conflicts occur and how they specifically affect performance in various tasks could strengthen the paper's contributions.

2. The applicability of the Multimodal Knowledge Consistency Fine-tuning method is discussed in the context of document understanding but may require further discussion for broader multimodal tasks. The paper could benefit from exploring how well these findings generalize to other domains beyond document understanding, such as scene understanding or visual reasoning.

3. The paper does not explore whether C&P conflicts can be mitigated through simple in-context learning strategies, such as performing VQA followed by OCR based on the previous context, or vice versa. This approach might provide insights into whether the conflicts are resolvable with less complex interventions.

4. The paper treats C&P conflict primarily as a consistency issue. However, it does not provide statistics on cases where the model's output consistency is low due to a failure to follow instructions correctly. Differentiating between consistency issues and instruction-following issues could offer a more nuanced understanding of the conflicts.

5. Ablation Study Metrics: The ablation study in Table 5 focuses on consistency metrics but does not include corresponding accuracy or F1 scores. Providing these metrics would offer a more comprehensive view of the method's performance, especially in terms of the trade-offs between consistency and accuracy.

**Questions:**

1. Can C&P conflicts be addressed or mitigated through simple in-context learning? For example, first performing VQA (OCR) and then conducting OCR (VQA) based on the previous context.
2. Is C&P conflict entirely a consistency issue? If the model does not follow the current instructions, the output consistency will naturally be lower. Could the authors provide statistics on the proportion of such cases?
3. The ablation study in Table 5 only presents the consistency metrics without corresponding accuracy or F1 scores. Could the authors provide these relevant metrics?
4. On tasks where the model already performs well, does C&P fine-tuning lead to a decline in performance?

---

> ### Author Response · Authors · 2024-11-21
>
> Thank you for dedicating your time to reviewing our paper and offering insightful feedback. We appreciate your recognition of the novelty of C&P knowledge conflicts. We have carefully considered your concerns and addressed them in the following response.
>
>
> > Insufficient Experimental Results on C&P Conflict Causes and Impact: The paper does not provide extensive experimental results on the root causes of C&P conflicts or a detailed analysis of their impact on downstream tasks. While it demonstrates the existence of these conflicts and the effectiveness of the proposed method in reducing them, a deeper investigation into why these conflicts occur and how they specifically affect performance in various tasks could strengthen the paper's contributions.
>
> Thank you for your valuable advice. Since the primary phase for knowledge acquisition predominantly occurs during the pre-training stage [1][2], we believe that **the root causes of C&P knowledge conflicts are related to the disconnection between cognition and perception during MLLM pre-training**. For example, in Qwen-VL-Chat, the VQA and OCR tasks during pre-training use two entirely different datasets [3], meaning that no single image is simultaneously trained for both tasks. This leads to the model struggling to establish an intrinsic connection between cognitive and perceptual knowledge. We will include an in-depth analysis in our revision.
>
> Regarding the impact of C&P knowledge conflicts, firstly, C&P knowledge conflicts undermine the explainability of MLLM responses (line 70), as these conflicts challenge the intuitive notion that cognition is consistent with perception and may affect users' trust in MLLMs. Furthermore, C&P knowledge conflicts also affect model performance. For instance, there is a situation (as shown in Figure 4b) where the model performs correctly on the OCR task but incorrectly on the VQA task. This indicates that while the model can accurately "see" the answer text, **it fails to fully utilize the perceptual knowledge in cognitive tasks**, resulting in incorrect answers. Statistical results show that, after fine-tuning with our method, 33.2% of such cases in Qwen-VL-Chat produce correct VQA answers. Therefore, we believe that addressing C&P knowledge conflicts may provide a potential direction for further improving the capabilities of MLLMs.
>
>
>
> > The applicability of the Multimodal Knowledge Consistency Fine-tuning method is discussed in the context of document understanding but may require further discussion for broader multimodal tasks. The paper could benefit from exploring how well these findings generalize to other domains beyond document understanding, such as scene understanding or visual reasoning.
>
>
>
> We acknowledge that exploring broader tasks enhances our paper. It is worth emphasizing that document understanding is a rapidly expanding research area with growing demand in industry. Additionally, to the best of our knowledge, we are **the first to identify and introduce the concept of C&P knowledge conflicts**, which affect even leading MLLMs. We believe document understanding is an ideal domain to demonstrate C&P knowledge conflicts. Moreover, our evaluation and fine-tuning methods **can be readily applied to other multimodal tasks**, such as using object detection as the perceptual task and general VQA as the cognitive task in scene understanding. As mentioned in Section 7, we intend to investigate this direction in future work.

---

> > ### Author Response · Authors · 2024-11-21
> >
> > > The paper does not explore whether C&P conflicts can be mitigated through simple in-context learning strategies, such as performing VQA followed by OCR based on the previous context, or vice versa. This approach might provide insights into whether the conflicts are resolvable with less complex interventions.
> > >
> > > Can C&P conflicts be addressed or mitigated through simple in-context learning? For example, first performing VQA (OCR) and then conducting OCR (VQA) based on the previous context.
> >
> >
> >
> > Thank you for highlighting the potential of in-context learning (ICL) in reducing C&P knowledge conflicts. We believe that accurately answering a VQA requires accurate OCR of the relevant text areas. Therefore, we adopt a simple in-context learning strategy involving a multi-turn dialogue, where **the model first performs OCR and then executes VQA**. Using GPT-4o as an example, the experimental results are as follows, with all values representing C&P consistency as percentages:
> >
> > |  | DocVQA | DeepForm | KLC | FUNSD | ChartQA | WTQ | **Average** |
> > | --- | --- | --- | --- | --- | --- | --- | --- |
> > | GPT-4o-ICL | 80.68 | 36.25 | 90.26 | 83.41 | 54.63 | 54.90 | **69.44** |
> > | GPT-4o | 75.59 | 16.71 | 79.84 | 65.61 | 63.86 | 55.83 | **65.51** |
> >
> >
> > Notably, during the rebuttal period,  GPT-4o's performance differs from the results of our September experiments as reported in the paper, possibly due to updates made to GPT-4o during this time. Therefore, to ensure a fair comparison, we re-conduct the GPT-4o experiments, as presented in the table above.
> >
> > The experimental results confirm the potential of in-context learning to reduce C&P knowledge conflicts. However, even with in-context learning, **GPT-4o still faces challenges in C&P knowledge conflicts**, achieving a C&P consistency of only 69.44%. We will continue to explore this direction in the future.
> >
> >
> >
> > > The paper treats C&P conflict primarily as a consistency issue. However, it does not provide statistics on cases where the model's output consistency is low due to a failure to follow instructions correctly. Differentiating between consistency issues and instruction-following issues could offer a more nuanced understanding of the conflicts.
> > >
> > > Is C&P conflict entirely a consistency issue? If the model does not follow the current instructions, the output consistency will naturally be lower. Could the authors provide statistics on the proportion of such cases?
> >
> >
> > We appreciate your comment on the instruction-following issues of MLLMs. We emphasize that **VQA and OCR are fundamental tasks for current MLLMs** [4], which are fully equipped to handle instructions related to these tasks. The prompts used for evaluation are carefully crafted based on the models' official guidelines, minimizing the risk of instruction-following failures. Furthermore, following your suggestion, we conduct **a human evaluation on instruction-following** by randomly selecting 100 samples from the experiments in Table 2, using GPT-4o and Qwen-VL-Chat as examples. The evaluation results show that GPT-4o fails to follow instructions in 1% of the samples and Qwen-VL-Chat in 2%, with failures referring to cases of refusal to answer or producing garbled output. We will provide additional details about the prompts and include a supplementary analysis in the revision.

---

> > > ### Author Response · Authors · 2024-11-21
> > >
> > > > Ablation Study Metrics: The ablation study in Table 5 focuses on consistency metrics but does not include corresponding accuracy or F1 scores. Providing these metrics would offer a more comprehensive view of the method's performance, especially in terms of the trade-offs between consistency and accuracy.
> > > >
> > >
> > > Thank you for your valuable suggestion. We recognize that including task performance (e.g., accuracy) from the ablation study offers a more comprehensive view of our method. Therefore, we present the following results, with **bold** indicating the best results and _italic_ indicating the second-best results.
> > >
> > > The following table shows the performance on the cognitive task (VQA) using the same metrics as Table 6:
> > >
> > > | Per. | Cog. | Conn. | DocVQA | DeepForm | KLC | FUNSD | ChartQA | WTQ |
> > > | --- | --- | --- | --- | --- | --- | --- | --- | --- |
> > > |  |  |  | 62.47 | 4.221 | 47.11 | 47.50 | 63.53 | 22.43 |
> > > |  | √ | √ | 63.33 | 34.06 | **65.54** | **52.44** | 64.13 | 23.00 |
> > > | √ |  | √ | 62.77 | 34.35 | _64.46_ | 51.02 | 62.93 | 23.74 |
> > > | √ | √ |  | **63.75** | **35.08** | 64.32 | _51.64_ | **64.45** | _24.59_ |
> > > | √ | √ | √ | _63.49_ | _34.35_ | 63.61 | 50.31 | _64.33_ | **24.68** |
> > >
> > > The following table shows the performance on the perceptual task (OCR) evaluated by ANLS:
> > >
> > > | Per. | Cog. | Conn. | DocVQA | DeepForm | KLC | FUNSD | ChartQA | WTQ |
> > > | --- | --- | --- | --- | --- | --- | --- | --- | --- |
> > > |  |  |  | 22.67 | 9.071 | 48.60 | 11.05 | 27.16 | 11.49 |
> > > |  | √ | √ | 45.97 | 29.67 | 71.76 | 32.12 | 84.07 | 49.46 |
> > > | √ |  | √ | **74.76** | _66.14_ | **90.16** | _62.71_ | **96.64** | **76.96** |
> > > | √ | √ |  | 74.11 | 65.49 | 87.77 | **62.85** | 96.30 | 74.78 |
> > > | √ | √ | √ | _74.21_ | **66.73** | _89.18_ | 61.99 | _96.62_ | _76.51_ |
> > >
> > > The model, employing complete multimodal knowledge consistency fine-tuning, achieves at least the second-best performance on most datasets. This demonstrates a trade-off between consistency and accuracy. However, **enhancing consistency does not significantly compromise accuracy**. We will include more relevant explanations in the revision.
> > >
> > >
> > >
> > > > On tasks where the model already performs well, does C&P fine-tuning lead to a decline in performance?
> > > >
> > >
> > > According to Table 6, InternVL2-2B and InternVL2-8B show minor declines on a few datasets where they originally performed well.  We attribute this to the possibility that both cognitive and perceptual responses may occasionally fail simultaneously while maintaining consistency, as illustrated in Figure 4a. However, our analysis indicates that such cases are rare. Moreover, considering the significant improvement in C&P consistency after fine-tuning, these "trade-offs" are acceptable.
> > >
> > >
> > >
> > > Again, we would like to express our gratitude for your careful evaluation of our paper, and we look forward to your further response.
> > >
> > >
> > >
> > > ### Reference
> > > [1] Zhou, Chunting, et al. "Lima: Less is more for alignment." Advances in Neural Information Processing Systems 36 (2024).
> > >
> > > [2] Kaddour, Jean, et al. "Challenges and applications of large language models." arXiv preprint arXiv:2307.10169 (2023)
> > >
> > > [3] Bai, Jinze, et al. "Qwen-vl: A frontier large vision-language model with versatile abilities." arXiv preprint arXiv:2308.12966 (2023).
> > >
> > > [4] Yin, Shukang, et al. "A survey on multimodal large language models." National Science Review (2024): nwae403.

---

> > > > ### Comment · Reviewer_kooV · 2024-11-27
> > > >
> > > > I appreciate the authors for supplemented experiments and additional responses. I do agree that C&P knowledge conflicts are interesting and crucial topics worth addressing. However, constraining the scope to document understanding significantly harms the dataset's comprehensiveness, leading to the work's limited impact. I would expect a further enhancement on the scope of the dataset and this work from a more general perspective. The current stage of work, however, is slightly below the acceptance standard of ICLR.

---

### Note · Authors · 2025-01-13

I have read and agree with the venue's withdrawal policy on behalf of myself and my co-authors.